# Proteomic Research in Peritoneal Dialysis

**DOI:** 10.3390/ijms21155489

**Published:** 2020-07-31

**Authors:** Mario Bonomini, Francesc E. Borras, Maribel Troya-Saborido, Laura Carreras-Planella, Lorenzo Di Liberato, Arduino Arduini

**Affiliations:** 1Nephrology and Dialysis Unit, Department of Medicine, G. d’Annunzio University, Chieti-Pescara, SS. Annunziata Hospital, Via dei Vestini, 66013 Chieti, Italy; lorenzo.diliberato@asl2abruzzo.it; 2Nephrology Department, Campus Can Ruti, Germans Trias i Pujol Research Institute (IGTP), REMAR-IGTP Group, Germans Trias i Pujol University Hospital, Carretera de Can Ruti, Camí de les Escoles s/n, 08916 Barcelona, Spain; feborras@igtp.cat (F.E.B.); mitroya.germanstrias@gencat.cat (M.T.-S.); lcarreras@igtp.cat (L.C.-P.); 3Department of Research and Development, CoreQuest Sagl, Tecnopolo, 6934 Bioggio, Switzerland; a.arduini@corequest.ch

**Keywords:** end-stage renal disease, peritoneum, peritoneal dialysis, peritoneal dialysis effluent, proteomics, biomarker

## Abstract

Peritoneal dialysis (PD) is an established home care, cost-effective renal replacement therapy (RRT), which offers several advantages over the most used dialysis modality, hemodialysis. Despite its potential benefits, however, PD is an under-prescribed method of treating uremic patients. Infectious complications (primarily peritonitis) and bio-incompatibility of PD solutions are the main contributors to PD drop-out, due to their potential for altering the functional and anatomical integrity of the peritoneal membrane. To improve the clinical outcome of PD, there is a need for biomarkers to identify patients at risk of PD-related complications and to guide personalized interventions. Several recent studies have shown that proteomic investigation may be a powerful tool in the prediction, early diagnosis, prognostic assessment, and therapeutic monitoring of patients on PD. Indeed, analysis of the proteome present in PD effluent has uncovered several proteins involved in inflammation and pro-fibrotic insult, in encapsulating peritoneal sclerosis, or even in detecting early changes before any measurable modifications occur in the traditional clinical parameters used to evaluate PD efficacy. We here review the proteomic studies conducted thus far, addressing the potential use of such omics methodology in identifying potential new biomarkers of the peritoneal membrane welfare in relation to dialytic prescription and adequacy.

## 1. Introduction

End-stage renal disease (ESRD) requiring RRT represents a growing problem worldwide and carries a significant economic burden [1]. Although kidney transplantation is the best treatment option, it is limited by the scarcity of donors. Thus, the majority of ESRD patients are treated by dialysis therapy, either hemodialysis or peritoneal dialysis (PD).

The peritoneum is a semipermeable membrane that performs several functions [2] and that can be used as a biological dialysis membrane [3]. The blood depuration mode (dialytic exchange) of PD is based on the exchange of solutes and removal of fluid from the blood (peritoneal capillaries) with a two liters (the usual volume) PD solution infused into the peritoneal cavity via an implanted intra-abdominal catheter. The effluent is then drained after a dwell time (4 to 8 h) before fresh dialysate is re-infused. This can be performed manually (continuous ambulatory PD; CAPD) with an average of four exchanges daily or employing a cycler (automated PD; APD) during the night (on average 8 h). The PD solution composition includes physiological concentrations of chloride, calcium, sodium, and magnesium, as well as a buffer (lactate and/or bicarbonate) and an osmotic agent in order to remove the daily excess fluid from the patient (peritoneal ultrafiltration (UF)). Glucose (molecular weight 180 Da) is the standard osmotic agent used, due to its efficiency, low cost, and acceptable safety profile. The glucose content in PD fluid is 10- to 50-fold higher than physiological serum concentrations, which creates an osmotic gradient allowing for removal of water, electrolytes, and toxins by UF-associated convection [4].

PD is an established home care, cost-effective RRT, which offers several advantages over the most commonly used dialysis modality, hemodialysis, including a more gradual and continuous solute and fluid clearance, minimal cardiac stress, better preservation of residual renal function, and similar survival [5].

Despite its potential benefits, PD is prescribed to only a minority of dialysis patients, about 10% in the USA and 13% in Europe [5,6]. To a significant extent, this discrepancy can be explained by major limitations regarding PD efficiency and sustainability [4]. In the short- and mid-term, infectious complications (primarily peritonitis) and catheter problems are the main reasons why this technique gets abandoned. For patients on long-term PD, bio-incompatibility of the dialysis fluid represents the main problem. Biocompatibility of a PD solution can be defined as its capacity to leave the natural anatomical and functional characteristics unmodified in time; and it can be divided into local (peritoneum cavity) and systemic. Currently, it is generally accepted that conventional PD fluids alter the anatomical and functional integrity of the peritoneal membrane (PM) over time by causing inflammation, neoangiogenesis, and fibrosis [7,8]. The progressive damage to the PM is evidenced by a faster peritoneal solute transport rate and by a decline in UF capacity, eventually resulting in UF failure, which is the main reason for technique failure [9]. Although several potential factors have been claimed as responsible for the poor biocompatibility of PD solutions [10,11], glucose is thought to be the main culprit. Besides playing a distinct role in the longitudinal changes to the structure and function of the PM, excessive intraperitoneal absorption of glucose from the dialysis fluid also has many potential systemic metabolic effects, including insulin resistance, new onset diabetes, and cardiovascular disease [12,13]. Strategies devised to reduce/eliminate glucose-associated toxicity (glucose sparing) represent one of the prime objectives in PD therapy [14,15].

Currently, the approaches to PD patient monitoring are mostly limited to approximating the delivered doses of dialysis and to measuring the transport status of the PM. To improve the clinical outcomes of PD, there is an unmet need for biomarkers as tools to identify patients who are at risk of PD-related complications and to guide personalized interventions [16]. For this purpose, PD effluent (PDE) represents a clinically relevant sample because it contains biomolecules that are indicative of the peritoneal health, peritoneal transport status, and ongoing pathological processes [16].

In this article, we reviewed the proteomic studies that have searched for biomarkers, enabling the diagnosis, prognosis, and therapeutic monitoring of pathological events related to PD.

## 2. Proteomics

Analysis of the protein content of a given sample, fluid, or tissue started back in the 19th century with the definition itself of proteins, although analyses of specific proteins had to wait until further scientific advances occurred. Among these, the discovery of the secondary structure of proteins by Pauling [17] and the development of spectroscopy methods to probe protein structure directly impacted research on the “proteome” as we currently know it.

The amino acid composition of proteins determines their specific migration in gel. Taking advantage of this essential characteristic, the initial proteomic studies were based on two-dimension gels that were able to districate complex mixtures of proteins. Since these early assays, in the last 20 years we have witnessed further developments in mass spectrometry (MS), microarrays, and protein chips that have enabled protein composition analysis of complex samples, and have profoundly changed our capacity to study the interaction among proteins, new drug design, the development of protein networks, and also the identification of biomarkers.

Of the various different methodologies, MS-based methods and micro arrays are probably the best choice technologies for the large-scale study of the proteome in a given sample. To begin with, MS-based proteomics studies were focused on the definition of proteins found in different species or tissues. Later, proteomics evolved to protein quantitation, based on the identification of so-called ”proteotypic” peptides of specific proteins [18]. Today, the application of MS-based proteomic techniques has led to wide definition of the human proteome. At the same time, new developments in targeted proteomics have introduced the possibility of quantitatively following up specific protein targets in a hypothesis-driven experiment [19].

In the workflow geared to biomarker discovery, whole-proteome profiling typically relies on non-targeted liquid chromatography–mass spectrometry (LC–MS), which defines a broad spectrum of the protein content of a given sample. Such an approach produces results of limited accuracy, which are usually presented as fold-change or relative up–down regulation of a given protein or group of proteins. After defining this “broad spectrum”, a second step may use targeted LC–MS techniques to specifically quantify a select group of proteins of particular interest as identified in the first approach, thus permitting a much more precise application in biomedical research [20].

The huge quantity of data generated by the above-mentioned methodologies requires the participation of bioinformatics and the use of specific software to analyze the data. It has also led to the appearance of international databases such as UniProt Knowledgebase [21], PeptideAtlas [22], or Proteomics Identifications Database [23], which require collaboration by multiple researchers, keeping these databases updated with their contributions and new findings. Having such a wide spectrum of protein changes and altered networks from a sample allows one to define “profiles” rather than “single biomarkers”, alerting to a pathological situation. While this is in fact of enormous interest for basic research and innovation departments, we should bear in mind that proteomic techniques are still expensive and technically complex (both to perform and to interpret). This precludes any rapid incorporation of this technology into routine clinical practice in hospitals. Thus, while using proteomics has meant a substantial change in basic research, it is still of paramount importance that we translate such a huge quantity of data into a manageable piece of information to be used at the direct point of care. For this purpose, conventional antibody-based techniques such as ELISA or immuno-chromatography and/or newly generated chip-based detection assays, which are designed to specifically evaluate single (or a few) interesting biomarkers, are probably easier to apply clinically.

As is the case in other fields of research and biomedicine, in PD, the enormous amount of information obtained from proteomic analysis of samples will allow for greater understanding and detection of changes in the PM. Undoubtedly, better knowledge of how the PM functions is crucial for the development of new therapeutic strategies and accurate formulation of dialytic fluids to increase the technique survival in PD patients. The European Training and Research in Peritoneal Dialysis (EuTRiPD) network accordingly considers “omics” technologies and bioinformatics–biostatistics to offer valuable tools for biomarker discovery, serving to improve patient management in PD [16].

## 3. Proteomic Investigations in Peritoneal Dialysis

In PD, since the dialysis solution continuously circulates through the abdominal cavity and peritoneal capillaries, PDE contains both proteins and peptides leaked from the blood, as well as those secreted by peritoneal cells and inflammatory cells. Before we proceed in the following sections to discuss the proteomic analysis of PDE in patients undergoing PD, it should be noted that uremia itself can cause structural and morphological changes to the peritoneum. At the time of PD catheter insertion, sub-mesothelial thickening and vasculopathy were observed in the peritoneum of uremic patients as compared to controls with normal renal function [7]. Wang et al. [24] examined the proteome of peritoneal fluids collected from ESRD patients at the time of catheter insertion as well as from subjects with normal renal function undergoing selective laparoscopic cholecystectomy. A total of 16 spots representing 12 proteins had altered expression levels among the peritoneal fluids of the study groups. Western blot analysis confirmed that in uremic samples—kininogen-1, apoptosis inhibitor 2, cat eye syndrome critical region protein 1, and apolipoprotein A-I—had higher expression levels, whereas synaptic vesicle 2-related protein, glial fibrillary acidic protein, and envelope glycoprotein C2-V5 region had lower expression. The increased expression may indicate a change in PM permeability to middle-sized proteins or peritoneal inflammation with proteins sloughing off, providing potential targets and mechanisms for uremic toxicity [24].

### 3.1. In Vitro Models of PD

Several studies in different in vitro models of PD have been performed in order to obtain insights into the mechanisms of PD-associated complications.

Kratochwill et al. [25] examined by proteomic and bioinformatic technologies the stress response of mesothelial cells to single and repeated exposure to an acidic lactate-based PDS. A reduction in biological processes in favor of reparative processes was found upon a single PDS exposure. Activated cellular responses included reparative mechanisms (protein metabolism, modification, and folding), signaling, carbohydrate metabolism, and inflammatory cell processes. Molecular chaperones represented a major subgroup of identified proteins. Repeated exposure to PDS was associated with a reduction in protein regulation, which is indicative of inhibition of stress response by elevated preinduced chaperones. This study [25] first describes elements of the reprogrammed proteome of mesothelial cells during recovery from exposure to PDS cytotoxicity. About 100 proteins proved to be significantly enhanced or diminished in abundance after full-PDF stress. These data were next evaluated in a functional analysis, with particular attention being paid to glucose-associated pathways, in order to examine the stress specifically triggered by glucose [26]. Four pathways were found to be associated with glucose metabolism. Interestingly, when mesothelial cells were exposed to glucose only at nonphysiologically conditions, changes of investigated glucose pathway-associated proteins were much lower when compared to full-PDS stress experiments [25]. These findings suggest that glucose exposure alone does not fully explain the differential abundance of these proteins, as well as the presence of additional factors in the activation of glucose-associated pathways [26].

Mesothelial cell damage by PDS leads to a complex cellular stress response with an important role for heat shock proteins, which are essential for repair and cytoprotection [27]. Proteomics and bioinformatics were used to search for the cellular pathways, which could alter the heat shock response in PDS-exposed mesothelial cells [27]. It was found that the release by mesothelial cells of danger signals caused, in turn, an elevated release of cytokines, associated with a sterile inflammation via interleukin-1 receptor-mediated pathways. Such sterile inflammation reduces cytoprotective chaperone expression and may accentuate the damage to mesothelial cells after PDS exposure. Thus, blocking the interleukin-1 receptor pathway might represent a useful way to limit damage in PD patients [27].

In a more recent study, Herzog et al. [28] performed a comparative proteomic analysis between young and senescent human mesothelial cells. Cellular senescence is a biological program initiated by various forms of stress, which is characterized by arrested cell growth and alterations in cell secretory phenotype, and that is of potential impact in PD. A total of 29 differentially abundant protein spots were found when the young and the senescent cell proteome were compared, with 11 proteins being identified (actin, cofilin-2, cytokeratin-7, transgelin-2, Hsp60, Hsc70, proteasome β-subunits, nucleoside diphosphate kinase A, and cytosolic 5′(3′) deoxyribonucleotidase). Changes in the senescent phenotype of mesothelial cells were mainly found in cytoskeleton-associated proteins, which may contribute to the abnormal morphology of senescent cells. Changes were also found in housekeeping and chaperone proteins, with potential functional impact on cellular stress response in PD [28]. Modification of mesothelial cells with O-linked-*N*-acetylglucosamine (O-Glc-NAc), a posttranslational protein modification relevant in cell survival, did not differ between young and senescent cells, though in senescent cells altered dynamics of O-Glc-NAc regulation were identified.

Finally, Kratochwill et al. [29] compared, via multiple protein expression and transcriptome data, the stress response of human omental peritoneal mesothelial cells exposed to PD solutions prepared in the laboratory and either heat-sterilized (glucose degradation product (GDP)-containing) or filter-sterilized (no GDPs). Differentially regulated protein abundance was grouped on the basis of correlating or noncorrelating transcripts. Heat-sterilized PDS was associated with 13 spots (seven containing heat shock proteins) characterized by less protein abundance but at the same time upregulation of mRNA expression. Whether this inverse relationship is due to a higher protein turnover or to translational inhibition or posttranslational modification remains to be established [29]. In contrast, a small group of proteins showed increased protein abundance and downregulated mRNA expression that could be indicative of a prolonged protein half-life by decreased degradation. Two further subgroups displayed concurrent regulation of protein and mRNA, suggesting regular translation of transcriptional regulation of gene expression into proteins. This study first introduces cross-omics technologies as a novel way to assess the bio-incompatibility of PDS, and evidences altered posttranslational regulation of heat shock protein expression by heat-sterilized PD fluid [29].

Although these studies might not directly relate to patient material, they definitely provide potential proteomic biomarkers that enable the diagnosis, prognosis, and therapeutic monitoring of pathological events related to PD.

### 3.2. Overview of PDE Proteome

The PM consists of three layers: the capillary endothelium, the interstitium, and the mesothelium. According to the three-pore model, which competently describes the function of PM, the capillary endothelium is the membrane’s main transport barrier [30]. Removal of retained solutes and excess water during PD exchange is of benefit to the patient. However, the removal is not selective, and unintended removal of useful substances can also occur.

In a representative overview of the PDE proteome, Raaijmakers et al. [31] used nano LC–MS/MS to identify 189 proteins in dialysates from 9 pediatric patients, with 84 proteins being shared by all patients. The majority of proteins derived from the extracellular matrix (84% vs. 11% plasma proteins), which indicates that proteomics could provide clinically relevant information about localization of and changes in proteins in the PM. Wang et al. [32] found proteomic differences between PDE from five diabetic patients and normal peritoneal fluids. After Western blotting confirmation, in diabetic samples, vitamin D-binding protein, haptoglobin, and alpha2-macroglobulin were detected at higher levels, whereas complement C4-A and immunoglobulin k were at lower levels. Loss (higher PDE levels) of proteins may be due to a change in the permeability of PM or leakage from peritoneal inflammation. Loss of vitamin D-binding protein could aggravate the symptoms of osteodystrophy, suggesting we should monitor vitamin D levels in serum of patients undergoing PD. The downregulation of C4-A in PDE may involve the defense system of peritoneal mesothelium and increased susceptibility to infection [32].

Diabetes mellitus is the most common cause of ESRD. The high glucose concentration of the disease may accelerate damage to the PM by increasing permeability and decreasing UF efficiency. Yang et al. [33] used proteomic technologies to examine dialysate samples obtained from diabetic (*n* = 12) and chronic glomerulonephritis (*n* = 12) PD patients as a way of noninvasively searching for evidence or predicting biomarkers of glucotoxicity-related damage to the peritoneum. A total of 10 identified proteins showed statistically significant differences between the two groups. In diabetic PDE samples, four proteins were upregulated (apolipoprotein A-IV, Zn-α2-glycoprotein, eukaryotic translation initiation factor 4A isoform 1, and human class I histocompatibility antigen), and six downregulated (albumin, alpha-1-microglobulin/bikunin preproprotein, apolipoprotein A-I, immunoglobulin G1 Fc fragment, mutant retinol-binding protein, and haptoglobin alpha2). This comparative proteomic study on two groups of dialysates may provide evidence that is useful for understanding the different peritoneal protein changes. Although these proteins may not be new biomarkers, they may nonetheless indicate a situation for possible drug treatment or glycemic control [33]. The same group analyzed seven patients suffering from chronic glomerulonephritis, whose PDE samples were collected at the beginning of CAPD treatment and after 1 year [34]. Proteins identified as immunoglobin (Ig) mu chain C region, fibrinogen gamma chain, and C-reactive protein showed higher levels in the early-stage sample, suggesting they appear predominantly in patients with chronic glomerulonephritis. In PDE samples obtained after 1 year of CAPD, higher levels of Ig delta chain C region, alpha-1-antitrypsin, histidine rich glycoprotein, apolipoprotein A-I, and serum amyloid P-component were found, which may result from initial peritoneal inflammation or changes in the permeability of the peritoneum to middle-sized proteins. Those proteins with higher levels in the peritoneal dialysate after 1 year of treatment may shed light on the mechanism of, or be a potential marker for, initial peritoneal damage in CAPD [34].

What progressively emerged from initial studies is that the proteome of PDE is quite complex. A major problem in the analysis of any complex proteome is the presence of highly abundant proteins such as albumin and immunoglobulins, which may restrict the identification of proteins with less abundant profiles and need to be removed before proteomic analysis. Development of an optimal sample preparation is a critical issue for the gel-based proteomic analysis of PDE. Zhang et al. [35] examined five different protocols for concentrating PDE samples: trichloroacetic acid in acetone, acetone, ethanol, ultrafiltration, and acetonitrile (ACN). PDE proteins precipitated with 75% ACN showed the greatest number of protein spots by 2-DE, with over 800 distinct spots. ACN selectively removes high-abundant proteins such as albumin (protein depletion), enriching samples with apolipoprotein-like proteins. Protein equalization is another way of deploying high abundance proteins, consisting of a selective partial removal of a protein or group of proteins, and hence enriching the immunoglobulins in the sample [36]. Oliveira et al. [37] used DL-dithiotreitol (DTT), a cheap and easy way to equalize the sample’s protein levels, before performing proteomic analysis of PDE from six patients on PD. DTT is usually applied to disrupt intramolecular and intermolecular disulphide bonds, helping to unfold proteins rich in such bonds and hence leading to their precipitation. Since the majority of the most abundant proteins are rich in disulphide bonds, they are preferentially depleted but not eliminated totally. A total of 49 spots were analyzed, revealing 25 proteins differentially expressed. Redundancy of the Ig kappa chain C region and fibrinogen beta chain was found, which might be linked to post-translational modifications or to the presence of proteases in PDE [37]. Biological pathways that were altered according to protein identification in PDE included the acute inflammatory response, response to wounding, response to stress, response to stimulus, regulation of immune system processes, and phospholipid efflux. It is interesting to note that all analyzed PDEs contained proteins involved in calcium regulation and metabolism such as fetuin-A, alpha-1-microglobulin/bikunin precursor, and vitamin D-binding protein. This finding might potentially explain the calcification of soft tissues in PD patients, an issue deserving further research but also suggesting the utility of PDE proteome analysis for understanding the pathophysiology of uremic complications.

The validity of protein equalization with DTT in removing most abundant proteins toward analysis of PDE proteome was subsequently confirmed [36,38]. Araùjo et al. [38] compared the performance of ACN and DTT-based methods over PDE samples as proteomics tools vis-à-vis analysis by 2D-GE and MS. Although the number and type of proteins proved different between the two approaches, annotation per gene ontology revealed the same biological pathways being affected. Most proteins identified in PDE were extracellular proteins involved in regulation processes through binding. Loss of some proteins including vitamin D-binding protein, modulators of the inflammatory response, and antioxidants (haptoglobin, ceruloplasmin) may have a negative impact on the PD patient. Removal of other proteins such as adipokine or proteins associated with inflammation and oxidative stress such as retinol binding protein 4 may be of benefit, however.

Combinatorial peptide ligand library (CPLL) technology is an efficient approach for decreasing the dynamic range concentration of complex protein mixtures and at the same time increasing the capture of previously undetected proteins. Each CPLL bead contains a unique hexapeptide ligand that interacts with one or a few proteins. Once the most abundant protein species have saturated their binding sites, the remaining molecules are washed away, whereas low abundant protein species are progressively enriched [39]. Lichtenauer et al. [40] established the CPLL system using artificial PD effluent and validated the approach by spike-in experiments. Using a combination of CPLL and tandem mass tag (TMT) multiplexing, Herzog et al. [41] examined PDE samples (*n* = 40 from 20 patients) collected in a previous randomized study comparing standard PD solution to AlaGn-containing PD solution [42]. The proteomic approach led to identification of 2506 PDE proteins, well beyond what was previously reported. To investigate the biological role of proteins widely differing from normal plasma in their abundance ranking in PDE, the authors selected the 100 least abundant proteins in PDE in relation to plasma concentrations (lower ranked), and the 100 most abundant proteins in PDE in relation to plasma concentrations (higher ranked). Among the lower ranked proteins, enrichment of the processes was found to be related to host defense, including neutrophil and platelet degradation, inflammation signaling, antimicrobial killing, regulation of stress response, and oxidative stress [41]. Among the higher ranked proteins, the processes were found to be enriched in relation to fiber formation and extracellular matrix formation, which is consistent with cell-derived signatures of mesothelial–mesenchymal transition [43].

Bruschi et al. [44] characterized, via the combined use of CPLL and 2-DE, PDE samples collected from 19 pediatric patients on APD treatment. As compared to the proteomic profile of untreated samples, CPLL-treated PDE samples were characterized by a large decrease in the five predominant proteins (albumin, immunoglobulins, alpha1-antitrypsin, serotransferrin, and alpha1-microglobulin), and by the detection of 724 new low abundant protein species [44]. A total of 29 protein spots showed differences in relation to APD vintage: 16 proteins were increased and 13 decreased in patients on long-term treatment. Proteins whose levels were modified were found to be involved in specific biological pathways such as those in response to wounding, complement activation, and immune response. However, only two proteins presented a quite reproducible trend that fits with a non-linear curve as a function of the duration of APD: gelsolin (decreased) and intelectin-1, which steadily increased. Low levels of gelsolin (a multifunctional actin-binding protein) are considered as reflective of consumption, providing an indirect marker of inflammation in a variety of clinical conditions [45]. In turn, intelectin-1 levels increase during acute infections. Thus, the simultaneous decrease of gelsolin and increase of intelectin-1 in PDE could provide a marker of chronic inflammation and progressive sclerosis [44]. Note that the biological role of these protein changes was confirmed in vitro by treating mesothelial cells with oxidative or pro-fibrotic stimuli. The study in question represents a major attempt to characterize biomarkers that could help to identify patients with subclinical inflammation and/or developing PM fibrosis.

### 3.3. Effects of Different PD Solutions on PDE Proteome

Glucose is the almost universally used osmotic agent in PD solution, at a concentration ranging from 1.5% to 4.25% (*w*/*v*), according to the patient’s ultrafiltration need. In five PD patients, Cuccurullo et al. [46] examined PDE samples obtained after the dialytic exchange with different glucose percentages in PD solutions (1.5%, 2.5%, or 4.25%). Quantitative differences between PD solutions were detected. Alpha1 antitrypsin, fibrinogen β chain, apolipoprotein A-IV, and transthyretin were found to be under-expressed with the highest osmolar solution (glucose 4.25%) as compared to the other solutions (1.5% and 2.5%). Since alpha-1 antitrypsin has been proven to be biologically active in PDE by inhibiting elastase activity and synthesis of platelet-activating factor [47], its lower expression in solutions with high glucose content could indicate the higher inflammatory potential of these solutions. In keeping with this is the lower expression of the fibrinogen beta chain, which could indicate a higher degradation rate in favor of fibrin, directly involved in inflammatory events [46]. With regard to transthyretin, it was found that as solution osmolarity increases, the monomeric form of the protein proportionally increases while concentration of the mature protein decreases. The transthyretin morphologic change could be involved in the inflammation process since monomeric transthyretin stimulates cytotoxic activity [48]. The fact that higher concentrations of glucose in PD solution may have unfavorable effects on the peritoneum was also shown in a study on 19 high-average transport CAPD patients [49]. Two types of PD solutions were evaluated (Stay-Safe Balance, Fresenius, Germany, *n* = 9 and Physioneal, Baxter, USA, *n* = 10), at both low (83 mM and 76 mM, Fresenius and Baxter solutions, respectively) and high (128 mM and 126 mM, respectively) glucose concentrations. PDE samples from high and from low glucose concentrations were collected consecutively after a 4-h stay in the peritoneal cavity. The authors found that regardless of the PD solution type, use of the higher glucose concentration was associated with a significantly increased loss of proteins—a total of ≈23% more proteins were removed with the Fresenius solution, and 18% with the Baxter solution. Moreover, increase in the glucose concentration of PD solution was associated with an augmentation of advanced glycosylation end products (AGEs) in PDE, which may reflect their levels in the serum and the tissues of the patient [49]. AGEs can cause significant injury in PD, since the chronic exposition of PM to high levels of AGEs resulting from the high glucose content in standard PD solutions results in fibrosis progression and damage to this membrane [49].

Given the toxicity of glucose contained in PD solution, several substances have been examined as alternatives to glucose as osmotic agents in PD. Only two such agents are currently used in glucose-free dialysate: icodextrin and amino acids. It should be noted, however, that both solutions only replace up to 50% of the daily glucose exposure [12]. Icodextrin is a glucose polymer derived from starch that is indicated for use during a simple long dwell per day, allowing for slow but sustained peritoneal ultrafiltration [50]. Icodextrin is of particular value in high transporter PD patients, and it improves patient UF without increased risk of adverse effects. A recent systematic review and meta-analysis from 11 randomized controlled trials, enriched with unpublished data from industry-sponsored and investigator-initiated studies, has shown that compared to a glucose-only PD regimen, use of icodextrin-containing PD solution has clear fluid benefits, such as improvement of peritoneal UF and fewer episodes of fluid overload [51]. In 16 pediatric patients on APD, Bruschi et al. [52] examined the proteomic profile (2DE and MS) of PDE after a 14-h daytime dwell, performed on two consecutive days with a 7.5% weight/volume icodextrin solution and a 3.86% weight/volume glucose solution. Removal of low-molecular-weight proteins such as beta2-microglobulin and cystatin C, taken as biomarkers of PD efficiency, was linear for both solutions and significantly higher with the icodextrin solution. Of 524 spots found in PDE with both solutions, 314 spots were significantly higher when using icodextrin. While removal of low molecular weight proteins, which are considered as uremic toxins [53], may be favorable to the PD patient, identification in icodextrin PDE of high molecular weight proteins (enzymes, transport proteins, and others) and of oxidized proteins may contribute to the development of clinical complications [52]. One important finding by this study is the linear correlation between total protein removal during dialysis with icodextrin or glucose in the same patient, which suggests that protein removal is a characteristic of any given subject undergoing PD and is a function of its peritoneum permeability.

To improve the bio-compatibility of PD solution, so-called biocompatible glucose-based dialysis solutions characterized by neutral or physiological pH and low content of GDPs using multi-chamber bags, have been introduced into the market [14]. However, although these solutions reduce in vitro damage to cells [54], they are also less potent inducers of beneficial cell stress and repair systems [9], which can result in chronic inflammation, complement activation, and increased vascularity [55]. Another approach may be that of using PD solution enriched with l-carnitine, which was proven to be more biocompatible than standard glucose-based PD solution in several experimental models [56]. Carnitine can also be combined with other compounds such as xylitol [57], a novel osmo-metabolic approach to formulating novel PD solutions [15], which is to be evaluated by clinical trials in an advanced state of development (NCT 04001036 and NCT 03994471).

Supplementation of the solution with alanyl-glutamine (AlaGn) could be another way to improve the biocompatibility of PD therapy. Extended exposure of mesothelial cells to a PD solution causes inadequate cellular stress responses, resulting in mesothelial cell vulnerability and decreased immune-competent cell function, which have been related to the low glutamine levels present in the peritoneal cavity during PD [58]. AlaGn is a glutamine-releasing dipeptide. Early clinical trials in PD patients showed that short-term supplementation of AlaGn in the PD solution restores stress response and improves cellular host defense in peritoneal cells [42,59], thereby confirming in vitro results [58]. A more recent double-blinded, randomized, crossover, phase II, proof-of-concept study in both CAPD and APD patients has confirmed the protective effects of AlaGn in terms of surrogate markers of peritoneal membrane status (appearance of CA125) and immune competence (ex vivo-stimulated interleukin-6 release), also involving reduced peritoneal protein loss [60].

Proteomic analysis can offer an attractive approach to understanding the molecular mode-of-action of AlaGn in the PD solution [61]. Use of AlaGn-enriched PD solution reduced the activity of mechanisms associated with PM injury and restored the biological processes involved in stress response and host defense [41]. Improvement in cellular stress response might occur by modulation of AKT-dependent pathways. Analysis of upstream regulating factors disclosed that the most strongly inhibited regulators were interferon gamma, vascular endothelial growth factor (VEGF), and transforming growth factor-beta1(TGF-β1) [41], indicating down-regulation of pro-fibrotic and pro-angiogenic processes [43]. Effects of AlaGn on regulating elements might point to a molecular explanation for the suppression of PM deterioration, as observed in animal models of chronic PD, showing reduced sub-mesothelial thickening and vascularization upon AlaGn supplementation in the PD solution [62]. The results of this novel and interesting study [41] support the potential of PDE proteomic analysis for defining pathomechanism-associated molecular signatures in PD.

### 3.4. Peritoneal Transport Characteristics

Solute transport through the PM is indicative of the efficacy of PD therapy. On the basis of the removal transport rate of solutes, which are small molecules, as measured in the peritoneal equilibration test (PET), PM can be classified as high, high average, low average, and low transporter [57]. PET yields three parameters: 4-h dialysate to plasma ratio of creatinine (D/P creatinine), 4- to 0-h dialysate glucose ratio, and 4-h ultrafiltration volume [63]. Peritoneal solute transport gradually increases with time on PD, and high solute transport is predictive of technique failure [9] and is associated with all-cause mortality and hospitalization rate [64].

In the first published proteomic analysis of PDE, Sritippayawan et al. [65] examined 20 PD patients with varying transport rates for possible differences in the PDE proteome. Five proteins were differentially expressed among the transport groups. As confirmed in a validation set of other patients using ELISA, increased levels of complement factor C4A and immunoglobulin kappa light chain were found in higher transport versus lower transport patients. This study, though preliminary, demonstrated that proteomic technology can be used to compare differences between different patient sub-groups. In a subsequent study, Wen et al. [66] used 2D-differentiated gel electrophoresis followed by quantitative analysis and found 10 protein spots with significantly different intensity levels among PD patients with different PM types. In particular, vitamin D-binding protein, complement C3, and apolipoprotein A-I displayed enhanced expression in PDE of high transporter patients, with no difference in their serum levels among different groups. Increased acute phase reactants such as complement C3 and apolipoprotein A-I in high transporters is in keeping with previous observations linking acute phase reactants to high and high average transport, and could suggest that the high transport status of PM could be linked to its inflammatory status.

More recently, a study (detailed in the following and peritonitis sections) on total protein *N*-glycomes in serum and PDE showed that the PM transport rate (D/P creatinine at PET test) was positively associated with triantennary glycans and the alpha2,6-syalilation of them, and negatively associated with diantennary glycans and the alpha2,6-syalilation of them in PDE, but not in serum [67]. The same glycan traits were found to be associated with the concentrations of VEGF and TGF-β1, which have been linked in PD to pathological changes to the peritoneum [68]. Higher alpha2,6-syalilation of diantennary glycans and a higher galactosylation of diantennary glycans in PDE were positively associated with cancer antigen 125, a marker believed to represent peritoneal cell mass [16], and thus could reflect a better clinical state [67].

### 3.5. Peritoneal Ultrafiltration Failure

As discussed above, the local noxious effects of glucose-only PD solutions lead over time to an increase of both peritoneal permeability and peritoneal surface area that from a functional standpoint translates into UF failure. On the other hand, the final clinical outcome is volume overload, the main complication in PD patients [69]. In order to establish the so-called true UF failure, “the rule of fours” has been proposed, whereby the net UF is less than 400 mL after a 4 h dwell using 4.25% glucose-only PD solution [70,71]. More commonly, however, UF failure is identified whenever the difference between the drained and the instilled PD solution, known as net UF, is less than expected. Four types UF failure have been described, although the most common kind of UF failure regards modifications of the PM associated with a significant increase in transport status, leading to rapid dissipation of the osmotic gradient and poor UF. These changes are triggered by the constant exposure of the PM to glucose, whereby the main physiopathological mediators are endothelial nitric oxide synthase (eNOS), TGF-β, VEGF, and complement activation [71]. Only a small percentage of PD patients suffer from early onset of UF failure, as the majority of them require a longer time of PD treatment (>2 years) along with a significant reduction in fluid removal from the kidney. Thus, the most common form of UF failure seems to be associated with a progressive vasculopathy, which may account for the vascular involvement of increased peritoneal permeability and peritoneal surface area [7,72].

In this regard, Bartosova et al. [73] analyzed the effects of chronic kidney disease (CKD) and PD on arteriolar transcriptomic and proteomic profiles in a pediatric population, chosen for being devoid of preexisting CVD as in adults, thereby allowing one to differentiate uremia- and PD-induced molecular arteriolopathic mechanisms. Omental arterioles for transcriptomic and proteomic analysis were isolated from children at the time of catheter insertion (CKD5 group; *n* = 21), children on PD with low-GDP, neutral pH solution (PD group; *n* = 21), and 13 children with normal renal function undergoing elective surgery. Findings were validated in other pediatric cohorts. Activation of metabolic processes in CKD5 arterioles and of inflammatory, immunologic, and stress–response cascades in PD arterioles, was detected by gene ontology analysis, with the highest upregulation of the complement system and respective regulatory pathways. These findings were concordant at the proteome level. In addition, a close correlation was found between dialytic glucose exposure and the degree of vasculopathy as well as to activation of complement and TGF-β pathways. The study in question was the first to provide evidence for an essential role by the activated arteriolar complement system and the TGF-β signaling cascade in PD-induced vasculopathy [73].

Vasculopathy is a common pathogenetic driver of CKD progression in diabetic patients. In an effort to better predict and formulate treatment to prevent and/or delay diabetic kidney function deterioration, recent studies have identified a strong association between type 2 diabetes and N-linked glycosylation of proteins [74,75]. In particular, the presence of glycosylated immunoglobulin G (IgG), glycosylation of which affects the inflammatory potential of IgG, was prospectively evaluated in the DiaGene study over a mean follow-up of 7 years in a large cohort of diabetic patients [75]. Thereafter, the association was evaluated between 58 IgG *N*-glycan profiles and the estimated glomerular filtration rate (eGFR) as well as albumin-to-creatinine ratio (ACR). This was because it is well established that IgG galactosylation, fucosylation and sialylation drive differential IgG function, ranging from inhibitory/anti-inflammatory to activating complement and promoting antibody-dependent cellular cytotoxicity [76]. In the DiaGene study, the pattern of IgG *N*-glycosylation, in particular, monosialylation, bisecting GlcNAc, and fucosylation with bisecting GlcNAc, indicated a pro-inflammatory state of IgG at baseline, coupled prospectively with a rapid deterioration in the kidney function of patients with type 2 diabetes. However, only eGFR correlated with the above IgG glycan pattern, as no significant associations were found between IgG glycan patterns and ACR. These findings seem to suggest that renal macroangiopathy rather than microvascular disease is coupled with the IgG *N*-glycosylation.

PD-related protein glycosylation has recently been evaluated in an embedded, open label multi-center prospective randomized clinical trial [67], in which PD patients were treated with standard glucose-only, lactate-buffered PD fluid, and either continued on the same PD fluid (group 1, *n* = 38), or switched to a glucose-only, bicarbonate/lactate-buffered PD fluid (group 2, *n* = 40). An additional third group of patients (group 3, *n* = 16) was not included in the randomized part of the study and had already been treated with the latter PD fluid before the study period. Serum and 24-h (overnight) dialysate samples were collected at 0, 6, 12, 18, and 24 months; derivatized; and had the glycome analyzed by MALDI-TOF–MS. In the total cohort of PD patients, a statistically significant increase in the pro-inflammatory pattern of glycated IgG (H3N5F1) was observed over time only in the PD dialysate and not in serum [67]. As expected, such agalactosylated IgG-related structures were also positively associated with peritonitis. Although the relative abundance of primarily IgG-derived glycans was not associated with several inflammation markers or PET-creatinine and PET-UF, no such association was tested for the abundance of pro-inflammatory, agalactosylated glycated IgG. In keeping with the relevance of preserving residual kidney function in maintaining euvolemia and delaying the occurrence of UF failure, both highly correlated with PD patient survival [77,78,79], it would be of great interest to investigate if the decline of residual kidney function and UF failure correlates with the pro-inflammatory and anti-inflammatory IgG *N*-glycosylation pattern in both diabetic and non-diabetic PD patients.

### 3.6. Peritonitis

Peritoneal infection remains one of the most common and serious complications of PD, since it may cause catheter removal and permanent transition to HD, as well as having a major impact on patient morbidity and mortality [5,80,81]. Moreover, peritonitis may contribute to the deterioration of the PM [82]. In the context of peritoneal infection, biomarkers are needed that can discriminate between infection and noninfectious inflammation [16]. It has been found that 20% to 25% of cultures for diagnosis of infection remain negative, despite the presence of clinical and biochemical signs of bacterial infection [83]. Culture-negative peritonitis may include episodes of sterile inflammation not requiring antimicrobial therapy. To improve the PD patient’s outcome, there is also the need for biomarkers predicting the risk of infections, as well as for biomarkers able to identify the risk of downstream complications, recurrent-relapsing infections, and even death [16].

In a preliminary study, Lin et al. [84] examined the PDE of 16 patients suffering from peritonitis, compared to PDE from 11 patients who had just started CAPD. Using a high-throughput profiling approach, beta2-microglobulin, a protein expressed on the surface of stimulated cells that may reflect immune activation, was proposed as a biomarker for PD-related peritonitis [84]. Tyan et al. [85] examined the proteome of PDE from 12 CAPD patients before/after peritonitis. Ten proteins differentially expressed between the two dialysates were identified, which were classified into three major functional groups: binding/transport, acute phase/immune response, and blood coagulation/hemostasis. In PDE, after the peritonitis episode, haptoglobin and antithrombin-III were proven to be upregulated, whereas a downregulation was found for heat shock 70 Da protein 1A/1B, apolipoprotein A-1, inter-alpha-trypsin inhibitor heavy chain H4 (ITIH4), fibrinogen gamma and beta chains, ceruloplasmin, alpha-1-antithripsin (AAT), and zinc-alpha-2-glycoprotein. Identification of downregulated acute phase response proteins (AAT, ITIH4) in PDE after an episode requiring therapy may provide a novel direct indicator for drug treatment of peritonitis [85]. Guo et al. [86] first applied a form of technology based on magnetic bead separation and MALDI-TOF–MS for screening proteins and peptides between 1-15 kDa mass ranges in PDE. This approach, which was fast, did not require sample concentration, and was sensitive, identified 15 PDE peaks statistically differing between patients with peritonitis and those peritonitis-free. On the basis of those peaks, a classification model was generated and validated in an independent testing set of 40 patients (20 with and 20 without peritonitis), showing 90.5% sensitivity and 94.7% specificity [86]. The technology used in this study, however, provided a poor representation of proteins <1 kDa and >15 kDa, which require a combination approach for maximum coverage of the PDE proteome.

Bacterial entry into the peritoneal cavity may occur via the dialysis catheter (skin microflora) and/or via translocation of intestinal microflora from the gut. Although improvements in PD technology and intensive patient training in aseptic technique have reduced the incidence of skin microbe-associated peritonitis, it remains unclear as to why certain PD patients are more susceptible than others to infection. Aldriwesh et al. [87] collected samples from nine patients at PD initiation and later in therapy in order to identify factors that may increase patient susceptibility to infection. They initially found that three bacteria (*Staphylococcus aureus*, *Staphylococcus epidermidis*, and *Klebsiella pneumoniae*), typically involved in peritoneal infections, grew in PDE with variability between patients but did not do so when incubated with the PD solution. Factors associated with the dwelling of PD solution within the peritoneal cavity making it more supportive of growth included reduced acidity, substantial residual glucose, nitrogen availability via protein release, and increased iron levels. Proteomic analysis identified the presence of transferrin in all PDE samples. Interestingly, analysis of the iron binding status of peritoneal transferrin found it to be iron-saturated, which indicates no further iron-limitation protection against pathogens such as that created by transferrin in the blood [88]. Upon using radioactive iron-labeled transferrin, peritoneal transferrin was found as a potential direct iron source for the growth of bacteria causing peritonitis [87]. Moreover, adrenalin and noradrenalin were found in PDE and may be involved in enhancement of bacteria growth via transferrin iron provision. This study suggests that the iron biology status of peritoneal dialysate may represent a risk factor for the development of infectious peritonitis, since iron availability is of major relevance for bacteria to establish an infection. Therapeutic implications may also follow, the authors having observed that restoring a more iron-restricted environment by addition of iron-free apo-transferrin to PD solution inhibited the growth of PD bacterial pathogens. This approach, which scavenges free iron and has been explored in various clinical conditions [88], might be of use in PD patients as an anti-infective prophylactic measure or adjuvant therapy in case of developed infectious peritonitis [87].

An innovative approach for biomarker identification in PDE—MS glycomics—was employed in a more recent study in 94 PD patients (80 CAPD, 14 APD) monitored for up to 24 months [67]. Protein *N*-glycosylation is a post-translational modification that influences protein function and is related to inflammatory process. Several changes in the glycosylation profile were found to be associated with PD-related complications [67]. The occurrence of peritonitis appeared to be related to a relative increase in IgG-related glycans and lower galactosylation of diantennary glycans. These *N*-glycan traits in PDE may reflect local peritoneal inflammation, since in serum they were not associated with peritonitis, and might potentially be candidate functional biomarkers of PD-associated peritonitis, deserving future study [67].

### 3.7. Encapsulating Peritoneal Sclerosis

Encapsulating peritoneal sclerosis (EPS) is a rare but potentially devastating disorder of PD. It is characterized by an acquired inflammatory fibrocollagenous membrane encasing the small intestine, which results in symptoms of bowel obstruction [89]. The prevalence of EPS in PD is between 0.4% and 8.9%, and the most significant risk factor for development of it is the duration of PD treatment [90]. At 8 years, about 10–20% of PD patients will develop EPS [90]. The pathogenesis of EPS remains poorly defined, but two hits seem to be required [91]. The “first” hit or predisposing condition is non-inflammatory peritoneal sclerosis resulting from repeated exposure to PD solution. A pro-inflammatory “second hit” precipitates a cascade of proinflammatory and proangiogenic cytokines [89]. Genetic variation in the receptor for AGEs might predispose to peritoneal deterioration [92]. Mortality approaches about 50% at 1 year after diagnosis of EPS in PD patients [90,93]. The rarity of the disorder, its variable presentation, and the lack of screening tools frequently lead to diagnosis being delayed.

Zavvos et al. [94] used multiple proteomic approaches to investigate for biomarkers able to predict the onset or confirm the diagnosis of EPS in PDE. The study was undertaken in prospectively collected PDE samples, and comparisons were made between patients diagnosed with EPS and controls matched for PD treatment exposure. Changes in collagen alpha1(1), gamma-actin, complement factors B and I, and anti-trypsin were found up to 3 to 5 years prior to EPS onset, while gelsolin, apolipoprotein A-II, apolipoprotein A-IV, and hemoglobin beta changed at least 2 years before onset. Late markers were also identified (orosomucoid-1, intellectin-1, alpha-2-HS-glycoprotein chain B) as being of potential diagnostic value when combined with symptoms. Most proteins identified in PDE samples from patients who subsequently developed EPS are consistent with their potential involvement in inflammation, the acute phase response, fibrogenesis, and extracellular matrix turnover [94]. Data from this study support the growing evidence that EPS is preceded by inflammatory and fibrotic peritoneal injury [95] and are worth further evaluation as candidate diagnostic/prognostic markers of EPS in PD therapy.

### 3.8. Extracellular Vesicles in PDE

The study of extracellular vesicles (EVs) has met with huge development and interest in recent years. It has been widely accepted that analysis of the EV composition from different body fluids may help to define new biomarkers of pathology and also contribute to the follow-up of patients in response to treatment. Plasma and serum are probably the most commonly used body fluids for proteomic analysis, but as EVs may be found in virtually any body fluid, many other sources of biomarkers have been investigated including urine, cerebro-spinal fluid, and peritoneal exudate of cancer patients, among others.

The time also came for analysis of EVs in peritoneal fluid from PD patients. In 2017, Akbari and co-workers [96] defined the presence of microparticles (MPs) in peritoneal fluid of PD patients. The authors suggested that these MPs were derived from mesothelial cells under stress conditions and accumulated in the peritoneal cavity during PD. It was also suggested that the assessment of MP levels in PDE—a waste product of PD—could be useful as a biomarker for PM damage. Of note, only MPs were analyzed in this study, while exosomes and other small vesicles and soluble factors were discarded from analysis. Again, in 2017, using size-exclusion chromatography (SEC) as the EV-enrichment methodology, a study by Carreras-Planella and co-workers [97] identified, characterized, and performed the first proteomic analyses of EVs derived from PDE (PDE-EVs). In contrast with the above-mentioned study, their results showed a faint though non-significant reduction in the number of EVs in PDE from longer-term PD patients (LTPs) as compared to newly enrolled patients (NEPs). Importantly, working from the MS results of EV samples, the authors described a set of well-conserved proteins among patients, mostly related to EVs, but also found that several proteins were significantly overexpressed in NEP compared to LTP (who showed quantitatively and qualitatively reduced expression), suggesting their possible use as biomarkers.

Only a few weeks later, Pearson et al. published another study on the proteomics of EVs and PD [98]. In this case, samples were segregated in terms of EV purification, as some of them were enriched by ultracentrifugation (UC) and some others by UC plus an additional size exclusion chromatography enrichment. As before, the results of gene ontology analysis confirmed the definition of an exosome enrichment component, and many of the proteins found in both studies were coincided. While both studies had the limitation of a small number of patients recruited (nine in [97] and eight patients in [98]), importantly both groups demonstrated that EVs could be isolated from peritoneal fluid, and that these EVs could be used as a source of biomarkers in a non-invasive fashion. Later, Corciulo and co-workers [99] described aquaporin (AQP1)-containing exosomes in PDE as biomarkers of dialysis efficiency. AQP-1 had also been found in samples from Pearson’s study [98]. On the basis of a study of 30 patients, the authors found that levels of AQP1 in PDE-EVs positively correlated with the efficiency of PD. The authors also used immunogold staining and transmission electron microscopy to show that AQP1 may be found to be associated with the surface of EVs. However, in this study, EVs were purified using UC alone, a technique that delivers EV samples with high levels of soluble non-EV contaminants [100].

Finally, a second study by Carreras-Planella et al. [101] recently described (for the first time in a longitudinal study) the evolution of the proteomic profile of PDE-EVs in PD patients. Aiming to compare the results of the PET assay with the proteomic data, they followed up 11 patients each semester for at least 18 months. At the endpoint of the follow-up, patients were classified as stable (no changes detected by PET) or unstable (changes in ultrafiltration capacity detected by PET). Note that as early as the 6-month follow-up, PDE-EV from the stable group showed significantly higher protein expression levels than the unstable patients. Importantly, PET changes indicating modified ultrafiltration capacity could only be detected 6 months later (12 months from the starting point). These results suggest that analysis of PDE-EV proteome may potentially be used to identify early biomarkers of PM alteration in PD patients. In that direction, the authors pointed to changes in the expression of several proteins, such as endoglin, as candidate biomarkers.

Although more research is still required, altogether these studies demonstrate the feasibility of studying the proteome of PDE-EVs as a non-invasive source of early biomarkers for the monitoring of PM alterations in PD patients. While proteomic approaches are still not feasible in the clinical setting, the identification of patterns or groups of candidate biomarkers using PDE waste fluids may positively impact the monitoring of PD patients. Agreement among different groups as to consistency in the method of collecting, harvesting, and analyzing EVs will undoubtedly help to achieve these objectives.

## 4. Conclusions

In PD therapy, the peritoneal membrane undergoes morphological and long-term functional alterations that limit the treatment and contribute to adverse patient outcome. To improve the clinical outcome of PD, there is a need for biomarkers to identify patients at risk of PD-related complications and to guide personalized interventions. Proteomics has emerged as one of the most attractive topics in disease biomarker discovery. Proteomic research in PD has shown its potential for developing valuable tools that improve patient management by enabling diagnosis, prognosis, and therapeutic monitoring of pathological events related to PD. Identification of proteins in PDE by proteomic technologies may yield information on specific phenotypes that are associated with peritoneal membrane function, risk of peritonitis, peritoneal membrane remodeling, or onset of encapsulating peritoneal sclerosis (Table 1).

However, despite the abundance of data, clinical translatability of proteomics is still lagging behind. Among other arguments against implementation of proteomics in daily clinical practice is the lack of guidance of how to place the proteomic biomarker in the clinical context. Some of the studies carried out thus far are wildly underpowered to allow identification of statistically robust clinical biomarkers, although are useful for proof-of-concept. Hence, there is the need for validation of these studies in an adequately powered, prospective, independent clinical trial including larger cohorts of patients.

The timely recognition of both local and systemic adverse effects connected with the use of PD may aid in the proper management of the patient and in technique survival. Proteomics is one such technology that could help to personalize treatment, to the potential benefit of ESRD patients on PD. A better understanding of the molecular mechanisms, whereby PD-associated toxicity is induced, will definitely provide better strategies to manage the side effects of this dialytic strategy without compromising its depurative benefit.

## Figures and Tables

**Table 1 ijms-21-05489-t001:** Proteomic strategy and main findings in peritoneal dialysis.

Proteomic Strategy	Condition	Findings	Reference
2DE followed by LC–MS/MS and WB	Uremia	Higher in uremia: KNG1, apoptosis inhibitor 2, CECR2, and APOA1.	[24]
2DE followed by LC–MS/MS	Diabetes mellitus	Higher in diabetic: DBP, HP, and B2M. Lower in diabetic: C4A and IgK.	[32]
RP-nano-UPLC–ESI–MS/MS followed by peptide fragmentation patterning	Diabetic vs. chronic glomerulonephritis	Upregulated in diabetic: APOA-IV, AZGP1, AIF4A1, and HLA-A. Downregulated in diabetic: albumin, AMBP, APOA-I, IgG1-Fc, mutant RBP, and HP alpha2.	[33]
2DE followed by LC–MS/MS	Chronic glomerulonephritis at the beginning of CAPD and after 1 year	Higher at the beginning: IGHM, FGG, and CRP. Higher after 1 year: IGHD, SERPINA1, HRG, APOA-I, and SAP.	[34]
ACN- and DTT-based methods before 2-D GE and MS	PD	Loss of DBP, HP, CP may be negative for PD. Removal of adipokine or RBP4 may be positive.	[38]
CPLL and 2DE	CPLL treatment in PDE from pediatric patients	Decrease in CPLL-treated samples: albumin, Ig, SERPINA1, TF, and A1M. Decrease along PD of GSN and increase of ITLN1.	[44]
1DE with nano-RP-HPLC–ESI–MS/MS and 2-DE with MALDI-TOF–MS	PD solutions at glucose 1.5%, 2.5%, or 4.25%	Under-expressed in 4.25% glucose: SERPINA1, FGB, APO A-IV, and TTR.	[46]
Multiple Affinity Removal LC Column-Human 6, 2DE DIGE, MS and 2D WB	Stay-Safe Balance vs. Physioneal solutions	Increase in higher glucose concentration: AGEs in PDE.	[49]
2DE and MS	7.5% icodextrin solution vs. 3.86% glucose solution	Higher removal of B2M and CST3 with 7.5% icodextrin solution.	[52]
1D immunoblot, 2D-DIGE, 2D WB, and saturation labeling	Standard PD solution vs. AlaGn-containing PD solution	AlaGln-containing solution reduced PM injury and improved cellular stress. Inhibition of upstream IFG, VEGF, and TGF-β1.	[42]
MALDI-Q-TOF–MS and MS/MS	Different transport rates	Increased in high transport: C4A, IGK.	[65]
2D DIGE and MALDI-TOF–MS/MS	Different PM types	Increased in high transport: DBP, C3, APOA1.	[66]
MALDI-TOF–MS and glycosylation profile	PM transport rate	Positively associated with triantennary glycans and the α2,6-syalilation of those, and negatively associated with diantennary glycans and the α2,6-syalilation.	[67]
LC–MS	CKD and PD on omental arterioles of pediatric patients	CKD: activation of metabolic processes. PD: inflammatory, immunologic, and stress-response cascades. Dialytic glucose correlates with PD vasculopathy and activation of TGF-β pathways. Activated complement system and TGF-β signaling cascade in PD vasculopathy.	[73]
Glycosylation profile	Type 2 diabetes	Different IgG *N*-glycosylation patterns in diabetes.	[75]
MALDI-TOF–MS and glycosylation profile	Different PD solutions over time	Increase of an IgG glycosylation pattern over time and in peritonitis.	[67]
2DE and SELDI-TOF–MS	Peritonitis	Increased in peritonitis: B2M.	[84]
2DE and RP-nano-HPLC–ESI–MS/MS	Peritonitis	Higher in peritonitis: HP, SERPINC1. Decreased in peritonitis: HSP70 1A/1B, APOA-1, ITIH4, FGG and FGB, CP, SERPINA1, and AZGP1.	[85]
Magnetic bead separation and MALDI-TOF–MS	Peritonitis	Different 1-15 kDa protein and peptide patterns.	[86]
MALDI-TOF–MS and radioactive iron-labeled transferrin	Peritonitis	Increased in peritonitis: iron-saturated transferrin. It can also act as bacteria growth source.	[87]
2D SDS-PAGE/MS and iTRAQ	EPS	3 to 5 years before EPS: changes in COL1A1, g-actin, CFB and CFI, and SERPINA1. 2 years before EPS: GSN, APOA2, APOA4, and HBB. With EPS symptoms: ORM, ITLN1, and AHSG chain B.	[94]
Immunogold staining and TEM	Dialysis efficiency	Exosomal AQP1 positively correlates with PD effluent and ultrafiltration, free water transport, and Na sieving.	[99]
LC–MS/MS	PM transport rate	Different extracellular vesicles proteome patterns upon PET.	[101]

2DE, two-dimensional gel electrophoresis; LC, liquid chromatography; MS, mass spectrometry; MS/MS, tandem MS; WB, Western blot; RP-nano-UPLC–ESI–MS/MS, reverse-phase nano-ultra performance liquid chromatography–electrospray ionization–tandem mass spectrometry; ACN, acetonitrile; DTT, DL-dithiotreitol; CPLL, combinatorial peptide ligand library; 1DE, one-dimensional gel electrophoresis; AGE, advanced glycosylation end products; RP-nano-HPLC–ESI–MS/MS, reverse phase nano-high performance liquid chromatography–electrospray ionization–tandem mass spectrometry; 2D-DIGE, two-dimensional differential gel electrophoresis; MALDI-(Q)-TOF, matrix-assisted laser desorption ionization (quadrupole) time-of-flight; SDS-PAGE, two-dimensional sodium dodecylsulfate polyacrylamide gel electrophoresis; iTRAQ, isobaric tagging for relative and absolute quantification; TEM, transmission electron microscopy; CAPD, continuous ambulatory peritoneal dialysis; CKD, chronic kidney disease; EPS, encapsulating peritoneal sclerosis; PET, peritoneal equilibrium test.

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
