# Peer review of "Proteomic Research in Peritoneal Dialysis"

_ijms, 2020, doi:10.3390/ijms21155489_

Round 1

Reviewer 1 Report

The manuscript reviewed proteomic researches conducted so far in peritoneal dialysis. Varied PDE biomarkers were identified in different pathological conditions caused by different PD solutions, peritonitis, encapsulating peritoneal sclerosis…etc. Identified PDE biomarkers are potentially to be useful tools for predicting peritoneal membrane function, risk of peritonitis, peritoneal membrane remodeling, or onset of encapsulating peritoneal sclerosis. Finally, the authors suggested that these identified biomarkers should be clinically validated with large cohorts of patients. The manuscript was well-prepared and thus I just have few minor comments.

Minor comments:

  1. It is recommended to provide a Figure illustrating all proteomic strategies used so far in peritoneal dialysis and which proteins were identified in each strategy.

  1. In the line 161, western blotting should be Western blotting (W should be capital).

  1. There is a typo at line 506, “N-glycosilation” should be “N-glycosylation”.

Author Response

1. It is recommended to provide a Figure illustrating all proteomic strategies used so far in peritoneal dialysis and which proteins were identified in each strategy.

We thank the Reviewer for her/his appreciation of our manuscript. Since it was difficult to summarize all the info in a figure, we have prepared a table (Table 1).

2. In the line 161, western blotting should be Western blotting (W should be capital).

Done, line 224.

3. There is a typo at line 506, “N-glycosilation” should be “N-glycosylation”.

Done, line 735.

Reviewer 2 Report

The review manuscript by Bonomini et al. discusses proteomic investigations in peritoneal dialysis (PD) in seven content sections, divided into an Overview of the PDE proteome, Effects of different PD solutions on the PDE proteome, Peritoneal transport characteristics, Peritoneal ultrafiltration failure, Peritonitis, Encapsulating peritoneal sclerosis and the role of Extracellular vesicles in PDE.

The text is comprehensively structured and well-written by renowned experts in the field and does definitely give a comprehensive overview of proteomics research. The individual papers are discussed in detail, providing elaborate biomedical insight into the individual studies.

Just two important points:

As my group has been a major contributor to exactly this research field, I would like to mention that there are yet undiscussed proteomics papers focusing specifically on in-vitro research in peritoneal dialysis, identifying key molecules and processes, such as the stress response of mesothelial cells in response to PD fluids (Kratochwill et al. JPR 2009 doi:10.1021/pr800916s), the role of glucose in PD fluids (Lechner et al. JPR 2010 doi:10.1021/pr9011574), sterile Inflammation in experimental PD (Kratochwill et al. Am J Pathol 2011 doi:10.1016/j.ajpath.2010.12.034), cellular senescence caused by PD fluid exposure (Herzog et al. BMRI 2015 doi:10.1155/2015/382652), and comparison of proteomics and transcriptomics of stress responses in mesothelial cells exposed to GDP-rich heat-sterilized versus GDP-free filter-sterilized PD fluids (Kratochwill et al. BMRI 2015 doi:10.1155/2015/628158). The authors could include these papers and potentially others exclusively focusing on proteomics and PD in vitro (either in the existing sections or in one that focuses on in-vitro models of PD). These studies might not directly relate to patient material but as long as they are dedicated PD research papers, they definitely provide potential proteomic biomarkers enabling the diagnosis, prognosis, and therapeutic monitoring of pathological events related to PD.

Regarding the CPLL technology I would like to stress that the approach was demonstrated by Lichtenauer et al. (Electrophoresis 2014 doi:10.1002/elps.201300499) half a year before the paper by Bruschi et al. in which the reference is missing. Lichtenauer et al. established the CPLL system using artificial PD effluent and validated the approach by spike-in experiments. Importantly, this assay was then used for the study by Herzog et al. (reference 54 in the manuscript) which consistently identified more than 2500 proteins in PDE using a combination of CPLL and TMT multiplexing. While there is definitely room for more than one approach to the PDE proteome, we showed in this comprehensive study that an appropriate bead-to-PDE ratio is crucial for overcoming the masking effects of high abundance proteins. Therefore, as some other studies are discussed in multiple sections, the authors should add the first description of the CPLL technique in PD research by Lichtenauer et al. and the deep PDE proteome characterization (current ref 54) to the section “Overview of the PDE proteome”.

The selection process of the discussed papers has not been described – which is fine for me, as some papers are a bit hard to find using general search terms. As far as I can judge, the selection of PD proteomics papers is complete (except the in-vitro aspect mentioned above). In the conclusion it could be mentioned (once more) that some of the studies while useful for proof-of-concept (such as the EV studies or other studies with less than 10 patients) are wildly underpowered to allow identification of statistically robust clinical biomarkers.

Author Response

As my group has been a major contributor to exactly this research field, I would like to mention that there are yet undiscussed proteomics papers focusing specifically on in-vitro research in peritoneal dialysis, identifying key molecules and processes, such as the stress response of mesothelial cells in response to PD fluids (Kratochwill et al. JPR 2009 doi:10.1021/pr800916s), the role of glucose in PD fluids (Lechner et al. JPR 2010 doi:10.1021/pr9011574), sterile Inflammation in experimental PD (Kratochwill et al. Am J Pathol 2011 doi:10.1016/j.ajpath.2010.12.034), cellular senescence caused by PD fluid exposure (Herzog et al. BMRI 2015 doi:10.1155/2015/382652), and comparison of proteomics and transcriptomics of stress responses in mesothelial cells exposed to GDP-rich heat-sterilized versus GDP-free filter-sterilized PD fluids (Kratochwill et al. BMRI 2015 doi:10.1155/2015/628158). The authors could include these papers and potentially others exclusively focusing on proteomics and PD in vitro (either in the existing sections or in one that focuses on in-vitro models of PD). These studies might not directly relate to patient material but as long as they are dedicated PD research papers, they definitely provide potential proteomic biomarkers enabling the diagnosis, prognosis, and therapeutic monitoring of pathological events related to PD.

We thank the reviewer for her/his positive evaluation of our manuscript. In the revised version, we have inserted a new section (3.1) on “In-vitro models of PD”, that includes all the studies suggested by the Reviewer.

Regarding the CPLL technology I would like to stress that the approach was demonstrated by Lichtenauer et al. (Electrophoresis 2014 doi:10.1002/elps.201300499) half a year before the paper by Bruschi et al. in which the reference is missing. Lichtenauer et al. established the CPLL system using artificial PD effluent and validated the approach by spike-in experiments. Importantly, this assay was then used for the study by Herzog et al. (reference 54 in the manuscript) which consistently identified more than 2500 proteins in PDE using a combination of CPLL and TMT multiplexing. While there is definitely room for more than one approach to the PDE proteome, we showed in this comprehensive study that an appropriate bead-to-PDE ratio is crucial for overcoming the masking effects of high abundance proteins. Therefore, as some other studies are discussed in multiple sections, the authors should add the first description of the CPLL technique in PD research by Lichtenauer et al. and the deep PDE proteome characterization (current ref 54) to the section “Overview of the PDE proteome”.

In the revised version, in the section”Overview of the PDE proteome” it has been inserted the study by Lichtenauer et al. (ref. 40), as well as part of the study by Herzog et al. (now ref. 41).

The selection process of the discussed papers has not been described – which is fine for me, as some papers are a bit hard to find using general search terms. As far as I can judge, the selection of PD proteomics papers is complete (except the in-vitro aspect mentioned above). In the conclusion it could be mentioned (once more) that some of the studies while useful for proof-of-concept (such as the EV studies or other studies with less than 10 patients) are wildly underpowered to allow identification of statistically robust clinical biomarkers.

As suggested, in the Conclusions it has been mentioned the limitation of some studies.

Reviewer 3 Report

Bonomini et al review the latest advanced in the field of proteomics applications in peritoneal dialysis. This subject is very interesting and of great interest to proteomics in general but specifically to the field of renal replacement therapy where it is urgent to identify a panel of biomarkers for early detection of peritoneal membrane failure. 

The paper is well written and covers all the major works developed recently in this field. There are a few details that should be considered to improve the overall quality and readability of the manuscript.

In the reviewer's point of view, the paper is too descriptive, and the manuscript would benefit from some artwork, figures. i.e. section 2 would benefit from a figure of the current strategies used in proteomics.

The conclusions of the manuscript seem very general and shallow, the authors are encouraged to significantly improve this section.

Author Response

In the reviewer's point of view, the paper is too descriptive, and the manuscript would benefit from some artwork, figures. i.e. section 2 would benefit from a figure of the current strategies used in proteomics.

We thank the Reviewer for her/his appreciation of our manuscript. We have added in the revised version a table (Table 1) dealing with current proteomic strategies and related identified proteins.

The conclusions of the manuscript seem very general and shallow, the authors are encouraged to significantly improve this section.

According to the referee’s suggestion, the Conclusions section has been largely revised.

Round 2

Reviewer 1 Report

The authors have amended the manuscript adequately and I have no further comments.

Reviewer 2 Report

I thank the authors for their revision of the manuscript. All my points have been adressed.